# Growth Hormone Upregulates Melanoma Drug Resistance and Migration via Melanoma-Derived Exosomes

**DOI:** 10.3390/cancers16152636

**Published:** 2024-07-24

**Authors:** Prateek Kulkarni, Reetobrata Basu, Taylor Bonn, Beckham Low, Nathaniel Mazurek, John J. Kopchick

**Affiliations:** 1Edison Biotechnology Institute, Ohio University, Athens, OH 45701, USA; pk585316@ohio.edu (P.K.); basu@ohio.edu (R.B.); tb919318@ohio.edu (T.B.); bl289321@ohio.edu (B.L.); nm282821@ohio.edu (N.M.); 2Molecular and Cellular Biology Program, Ohio University, Athens, OH 45701, USA; 3Department of Biological Sciences, Ohio University, Athens, OH 45701, USA; 4Department of Nutrition, Ohio University, Athens, OH 45701, USA; 5Environmental and Plant Biology, Ohio University, Athens, OH 45701, USA; 6Department of Biomedical Sciences, Ohio University, Athens, OH 45701, USA

**Keywords:** growth hormone, exosomes, melanoma, chemotherapy resistance, ABC transporters, N-cadherin, MMP2, pegvisomant

## Abstract

**Simple Summary:**

Melanoma, a severe type of skin cancer, often becomes resistant to chemotherapy, making it difficult to treat. This research investigates a novel mechanism by which growth hormone (GH) contributes to chemotherapy resistance. We examined small particles, called exosomes, which are released from melanoma cells treated with GH and found to carry proteins that increase drug resistance and cancer cell movement. The effects were more pronounced when GH was combined with the chemotherapy drug, doxorubicin. We also found that blocking the GH action with a drug called pegvisomant reduced the expression of these exosomal proteins, ultimately making the cancer cells more responsive to chemotherapy and less likely to migrate. Our findings provide new insights into how GH action promotes melanoma chemoresistance via exosomes, suggesting that targeting/inhibiting GH action could improve melanoma treatment.

**Abstract:**

Drug resistance in melanoma is a major hindrance in cancer therapy. Growth hormone (GH) plays a pivotal role in contributing to the resistance to chemotherapy. Knocking down or blocking the GH receptor has been shown to sensitize the tumor cells to chemotherapy. Extensive studies have demonstrated that exosomes, a subset of extracellular vesicles, play an important role in drug resistance by transferring key factors to sensitize cancer cells to chemotherapy. In this study, we explore how GH modulates exosomal cargoes from melanoma cells and their role in drug resistance. We treated the melanoma cells with GH, doxorubicin, and the GHR antagonist, pegvisomant, and analyzed the exosomes released. Additionally, we administered these exosomes to the recipient cells. The GH-treated melanoma cells released exosomes with elevated levels of ABC transporters (ABCC1 and ABCB1), N-cadherin, and MMP2, enhancing drug resistance and migration in the recipient cells. GHR antagonism reduced these exosomal levels, restoring drug sensitivity and attenuating migration. Overall, our findings highlight a novel role of GH in modulating exosomal cargoes that drive chemoresistance and metastasis in melanoma. This understanding provides insights into the mechanisms of GH in melanoma chemoresistance and suggests GHR antagonism as a potential therapy to overcome chemoresistance in melanoma treatment.

## 1. Introduction

Melanoma, the most aggressive form of skin cancer, remains a significant public health concern, with the American Cancer Society projecting approximately 100,640 new cases in the US in 2024 [1]. Recent advancements in diagnostic options and therapeutic interventions have raised the 5-year overall survival rate to an impressive 99%, but it significantly drops to 35% in the advanced stages of the disease. This drop in patient survival is mostly attributed to metastasis and resistance to existing therapies [2]. Understanding the complex mechanisms of tumor metastasis and drug resistance is crucial in developing groundbreaking treatments and improving patient outcomes. 

Chemoresistance in melanoma involves multiple factors, including the tumor microenvironment (TME), enhanced drug efflux, and an elevated rate of epithelial-to-mesenchymal transition (EMT) [3,4,5]. Exosomes, tiny extracellular vesicles (EVs) released from the cells, are being recognized as key players in mediating these processes [6,7,8]. Exosomes are a subset of EVs that range from 50–150 nm in size, serving as vehicles for the transfer of proteins, RNA, and lipids -- collectively referred to as cargo -- from one cell to another [9]. 

EMT is a fundamental cellular process that is integral to embryonic development and wound healing, and it plays a crucial role in tumor growth [10]. During EMT, cancer cells lose their epithelial characteristics and acquire enhanced tumor cell motility, facilitating metastasis. Extensive studies have shown that EMT is intrinsically linked to cancer therapy resistance [11,12,13]. The mechanisms include the conversion of tumor cells into tumor stem cells [14], the reduced expression of drug carriers essential for drug internalization [13], and the upregulation of EMT transcription factors by non-coding RNAs [15,16]. Notably, tumor-derived exosomes have been implicated in the transfer of drug efflux pumps and EMT markers to sensitive cancer cells, contributing to chemoresistance [17,18,19,20]. 

Growth hormone (GH), an anterior pituitary hormone, is crucial in regulating longitudinal growth, organ development, and whole-body metabolism, as well as diseases such as diabetes and cancer [21]. Remarkably, patients with excess GH secretion have an increased risk of multiple cancers, while patients with no GH action (Laron Syndrome) are protected from all types of cancer [22,23,24]. GH also exerts distinct autocrine and paracrine actions via multiple cells in TME [25,26,27]. Previously, we have identified the expression of the GH receptor (GHR) gene in multiple NCI-60 cell lines, with overexpression observed in melanoma [28]. GHR activation of these cells leads to diverse cellular effects, including the induction of EMT, senescence, the upregulation of multidrug efflux transporters, and the downregulation of apoptosis [26]. Importantly, we have shown that GH upregulates the expression of ATP-binding cassette-containing (ABC) transporters in melanoma and other cancers, which in turn, confers increased resistance to chemotherapy, both in vitro and in vivo [29,30,31,32]. Moreover, we and others have shown the crucial roles of GH in the induction of EMT in various cancers [33,34,35,36], including melanoma [37,38]. Several of these ABC transporters and EMT markers, mediated directly or indirectly by GH, are shown to be trafficked from drug-resistant cancer cells via exosomes, conferring drug resistance on the recipient cells [39,40]. 

Therefore, in the present work, we aimed to elucidate the effects of GH on exosomal cargoes from donor cells and their subsequent effects on recipient cells, all contributing to drug resistance. For this, we investigated the influence of GH on the drug efflux pumps, EMT markers, and matrix metalloproteinases (MMPs) implicated in tumoral chemoresistance [11,41,42]. Through this investigation, we sought to uncover novel insights into the molecular mechanism of GH action in driving chemoresistance via melanoma-derived exosomes.

## 2. Materials and Methods

Cell culture and treatments: The human melanoma cell lines, MALME-3M (HTB-64) and SK-MEL-28 (HTB-72), were obtained from the American Type Culture Collection (ATCC, Manassas, VA, USA), while the SK-MEL-30 cell line was acquired from Creative Bioarray (Shirley, NY, USA). Cells were grown and maintained in Iscove’s Modified Dulbecco’s Medium (IMDM), Eagle’s Minimum Essential Medium (EMEM), and Rosewell Park Memorial Institute (RPMI) media, respectively, supplemented with 10% fetal bovine serum (# 10-082-147, Thermo Fisher Scientific, Waltham, MA) and 100 U/mL penicillin–streptomycin (#15-140-22, Thermo Fisher Scientific, Waltham, MA, USA). Cells were grown in a humidified incubator at 37 °C and 5% CO_2_. Recombinant human GH (#ABIN2017921, Antibodies-online, Pottstown, PA, USA) at 50 ng/mL, doxorubicin at the EC50 dosage, and pegvisomant (Somavert, Pfizer) at 500 nM were added in the respective treatment media supplemented with 2% exosome-depleted fetal bovine serum (#EXO-FBS-250A-1, System Biosciences, Palo Alto, CA, USA). We confirmed that GH promoted and the GHR antagonist, pegvisomant, attenuated phosphorylation levels of STAT5 in the aforementioned treatment combinations in all the three melanoma cell lines (Appendix A). The EC50 values were determined as 0.7 µM for Malme-3M, 1.5 µM for SK-MEL-28, and 2.8 µM for SK-MEL-30 (Appendix A). The melanoma cells were treated with doxorubicin at the EC50 dosage for 96 h, with replacement every 48 h.Exosome isolation: The supernatants of the cell cultures from the respective treatments were centrifuged at 3000× *g* for 15 min at 4 °C to remove the cell debris. Further, the supernatant was passed through the 0.22 µm filter (Millipore Sigma, Burlington, MA, USA) to remove the relatively large vesicles. To effectively concentrate exosomes from large volumes, ultrafiltration was employed using Amicon Ultra 15 mL centrifugal filters (100 kDa NMWL, Millipore Sigma, Burlington, MA, USA) [43]. Next, the ExoQuick reagent was added to the supernatant in 1:5 ratio, according to the manufacturer’s instructions (Systems Biosciences, Palo Alto, CA, USA) and incubated overnight at 4 °C with no rotation. Following incubation, the samples were centrifuged at 1500× *g* for 30 min. The supernatant was aspirated, followed by a brief centrifugation step of 1500× *g* for 5 min to facilitate further removal of the supernatant. The final pellet was resuspended in phosphate-buffered saline (PBS) for the downstream analysis.Nanoparticle tracking analysis: Exosome labeling was conducted using the EV tracker green NTA labeling kit (Systems Biosciences, Palo Alto, CA, USA). Briefly, the pre-warmed reaction buffer was mixed with ExoGlowTM dye in a ratio of 5:1, and then 5 μL of the working solution was added to 200 μg of sample and thoroughly mixed by pipetting. The samples were incubated at room temperature for 30 min while protected from light. A microscopic analysis was performed using Zetaview (Particle Matrix, Germany), equipped with a 520 nm laser, a 550 nm long pass cutoff filter, and an sCOMS camera.Transmission electron microscopy: Exosome samples were fixed with 2% paraformaldehyde for a minimum of 2 h at 4 °C, followed by adsorption onto glow-charged copper grids coated with formvar–carbon (#FCF-Cu-50, Electron Microscopy Sciences, Hadfield, PA, USA) for 20 min. Subsequently, after washing with 0.1 M phosphate buffer, the bound exosomes were fixed with 1% glutaraldehyde for 5 min. After washing with distilled water, the samples were negatively stained with 1% uranyl acetate for 1 min. The grids were air-dried and imaged using a FEI Technai G2 Spirit transmission electron microscope (Thermo Fisher Scientific, Waltham, MA, USA) operating at 80 kV, employing a Macrofire digital camera (Optronics, Inc, Chelmsford, MA, USA) and AMT image capture software version 5.42 (Advanced Microscopy Techniques, Woburn, MA, USA).Protein extraction and western blot: Protein extraction and western blot were performed as described previously [29]. Briefly, protein extraction was performed using a 1X RIPA buffer (#R-0278, Sigma Aldrich, St. Louis, MO, USA) containing 1X HaltTM protease and a phosphatase inhibitor cocktail (#78442, Thermo Fisher Scientific, Waltham, MA, USA). The protein concentration was quantified using the Bradford assay (#B6916, Sigma Aldrich, St. Louis, MO, USA) and 30 μg of protein was loaded onto 4–16% gradient SDS-PAGE denaturing gels. Further, the proteins were transferred to the polyvinylidene fluoride membranes, blocked with 5% BSA solution in 1X TBST-T and probed using target-specific antibodies. The exosomal markers in the protein extracts from the Malme-3M exosomes were determined using antibodies specific for CD63, CD9, and CD81 (#EXOAB-CD63A-1, #EXOAB-CD9A-1, #EXOAB-CD81A-1 SBI, Palo Alto, CA, USA). To determine the ABC transporters, the EMT markers, and the MMPs, protein extracts from Malme-3M exosomes were determined using antibodies specific for ABCC1, ABCC2, ABCB1, ABCG2, N-cadherin, E-cadherin, MMP2, and MMP9 (#72202, #125595, #13342, #42078, #13116, #3195, #87809, #13667). β-actin (#4970, CST, Denver, MA, USA) was used as a loading control. For detection, anti-rabbit IgG, an HRP-linked secondary antibody (#7074, CST, Denver, MA, USA), and a SuperSignal West Femto Maximum Sensitivity Substrate (#34095, Thermo Fisher Scientific, Waltham, MA, USA) were used.RNA extraction and RT-qPCR: The RNA was extracted, and RT-qPCR was performed as previously described [44]. Briefly, the total RNA was extracted using an IBI Scientific total RNA extraction kit (Dubuque, IA, USA), following the manufacturer’s protocol. Up to 2000 ng of complementary DNA (cDNA) was synthesized from isolated exosomal RNA. Further, quantitative real-time polymerase chain reaction (qRT-PCR) was performed using Applied Biosystems reagents (Thermo Fisher Scientific, Waltham, MA, USA) following the manufacturer’s protocol. The primers used were GH (Forward: AGGAAACACAACAGAAATCC, Reverse: TTAGGAGGTCATAGACGTTG). The expression levels of differentially expressed RNAs were compared using the 2-ΔΔCT method. β-actin (Forward: GACGACATGGAGAAAATCTG, Reverse: ATGATCTGGGTCATCTTCTC) was used as an internal control for the RNA analysis.Cell migration assay: The cells were seeded at 30,000 cells per well in 12-well plates. After 24 h, a scratch wound was made using a 200 µL pipette tip along the midline of each well. The cultures were gently washed with PBS to remove the loose cells. The cells were maintained in the respective media with Exo^control^ (from PBS-treated cells), Exo^GH^ (from GH-treated cells), Exo^doxo^ (from doxorubicin-treated cells), Exo^GH+doxo^ (from cells treated with GH and doxorubicin), and Exo^GH+doxo+Peg^ (from cells treated with GH, doxorubicin, and pegvisomant) for 24 h. For each treatment, 20 μg/mL of exosomes were added [45,46,47]. The scratch area was imaged at the start and end of the assay using a BioTek citation-3 microplate imager (Gen5v2.09.2 software) and quantified using ImageJ software (version 1.8.0_345). Three individual experiments were performed.Drug retention assay: Melanoma cells were treated for 12 h with Exo^control^, Exo^GH^, Exo^doxo^, Exo^GH+doxo^, and Exo^GH+doxo+Peg^. On the day of the assay, the cells were trypsinized, counted, and suspended in cold DiOC2(3) dye on ice for 30 min (EMD Millipore, ECM 910). The cells were then centrifuged, the supernatant was removed, and the cell pellets were resuspended in a cold efflux buffer. The resuspended cells were distributed in equal parts with one set serving as the control and the other two parts kept in a 37 °C water bath for 20 min and 60 min, respectively. The cells were then washed, resuspended, and the cells’ suspension was dispended into the wells of a black-walled 96-well plate. Fluorescence was measured using the fluorescent BioTek citation-3 microplate imager (Gen5v2.09.2 software) at an excitation wavelength of 485 nm and an emission wavelength of 530 nm. Two individual experiments were performed for each cell line.Chemosensitivity assay: MALME-3M, SK-MEL-28, and SK-MEL-30 cells were seeded at 500 cells per 50 µL per well in a 96-well plate. After incubating for 24 h, the cells were treated with 20 µg/mL of Exocontrol, ExoGH, Exodoxo, ExoGH+doxo, and ExoGH+doxo+Peg. Twelve hours later, the cells were exposed to a dose titration of doxorubicin at the specified concentrations in 25 µL. The cell viability was assessed 72 h after doxorubicin treatment as previously described [48].Statistical Analysis: For all the experiments, the analysis was performed by one-way or two-way ANOVA with Tukey’s multiple comparison test using GraphPad Prism 8.0 (GraphPad Software). *p* < 0.05 (*), *p* < 0.01 (**), and *p* < 0.001 (***) were considered statistically significant.

## 3. Results

### 3.1. GHR Antagonism Suppresses Melanoma Exosome-Mediated Increase in Drug Efflux

The size and concentration quantification of the Malme-3M, SK-MEL-28, and SK-MEL-30 melanoma-derived exosomes demonstrated an average peak size at 107.1 nm and an average concentration of 1.5 x10^11^ particles/mL (Appendix A). The identification of all the exosomes derived from all three melanoma cell lines was further confirmed by the exosomal markers CD63, CD9, and CD81 using immunoblotting (Appendix A). Additionally, the morphological characteristics of the Malme-3M melanoma-derived exosomes were confirmed using transmission electron microscopy (Appendix A). 

GH has been shown to induce chemoresistance, both in vitro and in vivo, in melanoma [29], hepatocellular carcinoma [32], and pancreatic cancer [49], as shown by our laboratory. Blocking the GHR action attenuates this process, making the tumor cells drug sensitive. Recent studies revealed that drug-resistant cancer cells release exosomes that assist drug-sensitive cancer cells in acquiring chemoresistance [50,51,52]. To investigate if and how GH affects the transfer of chemoresistance via exosomes, we employed a drug-retention assay in recipient melanoma cells. 

For this study, we isolated exosomes from three melanoma cell lines: Malme-3M, SK-MEL-28, and SK-MEL30, each treated with the following conditions: either hGH or doxorubicin alone, hGH and doxorubicin, or a combination of hGH, doxorubicin, and pegvisomant (a GHR antagonist) [53,54,55]. The exosomes were referred to as Exo^control^ (from PBS-treated cells), Exo^GH^ (from GH-treated cells), Exo^doxo^ (from doxorubicin-treated cells), Exo^GH+doxo^ (from cells treated with GH and doxorubicin), and Exo^GH+doxo+Peg^ (from cells treated with GH, doxorubicin, and pegvisomant). These exosomes were then administered to treatment-naïve melanoma cells (recipient cells). We then administered DiOC2(3), a surrogate dye for chemotherapy, to conduct the drug-retention assay in the recipient cells. We measured the amount of dye retained as an indicator of the chemoresistance observed in the cells receiving the exosomes.

A significantly lower dye retention was observed in the Malme-3M cells (by 20%) administrated with Exo^GH^ as compared with the control (Figure 1A). Similarly lower, but not significant, dye retention was observed in the SK-MEL-28 or SK-MEL-30 cells as compared with the respective controls (Figure 1B,C). Upon administering Exo^doxo^, a significant decrease in drug retention was observed in the Malme-3M (by 10%) and SK-MEL-28 (by 20%) cells as compared with the controls. A similar lower drug retention, albeit not significant, dye retention was observed in all three cell lines when administered with Exo^GH+doxo^.

The dye retention levels were rescued to control levels in the three cell lines when administered with Exo^GH+doxo+Peg^ (Figure 1B–D). A notable enhancement in drug retention was noted in cells treated with Exo^GH+doxo+Peg^ compared with those treated solely with Exo^GH^. Specifically, the Malme-3M and SK-MEL28 cells exhibited a 35% and 32% increase in drug retention under this combined treatment regimen, respectively (Figure 1B,C). The dye retention was maintained in Malme-3M and exhibited an even more pronounced retention (47% and 41%) in both SK-MEL-28 and SK-MEL-30 at a later time point (60 min), respectively, suggesting prolonged and enhanced drug retention. 

Furthermore, to determine how the aforementioned exosomes affect the sensitivity of the recipient cells, we examined the IC50 values for doxorubicin following exosome administration. A distinct increase in IC50 was observed in the Malme-3M (by 2.6-fold), SK-MEL-28 (by 2-fold), and SK-MEL-30 (by 3.8-fold) cells administrated with Exo^GH^ as compared with the control (Figure 1D–F). A slight increase in the IC50 values was observed in the Malme-3M (by 1.6-fold) and SK-MEL-30 (by 3-fold) cells administrated with Exo^doxo^ as compared with the control (Figure 1D–F). A similar increase in the IC50 values was observed in the Malme-3M (by 2-fold) and SK-MEL-30 (by 3.8-fold) cells administrated with Exo^GH+Doxo^ as compared with the control (Figure 1D–F). Strikingly, the IC50 values were comparable with the controls in all three cell lines when administered with Exo^GH+doxo+Peg^. Interestingly, an evident reduction in drug retention was noted in the cells treated with Exo^GH+doxo+Peg^ compared with those treated solely with Exo^GH^. Specifically, the Malme-3M, SK-MEL-28, and SK-MEL-30 cells exhibited a 60%, 32%, and 63% decrease in IC50 under this combined treatment regimen, respectively. 

### 3.2. GH Elevates the Expression of ABC Transporters in Melanoma-Derived Exosomes and in Corresponding Recipient Cells

#### 3.2.1. Effects of GH on ABC Transporters in Melanoma-Derived Exosomes

As GH regulates ABC transporter (efflux pump) expression in tumor cells [29,31,32], we checked the ABC transporter levels within the released exosomes responsible for the modulation of drug retention in the recipient cells. These pumps have been characterized and studied as efficient facilitators of drug efflux from cancer cells and correlated with increased chemoresistance. Studies have shown that exosomes transfer multiple ABC transporters, thereby conferring chemoresistance in tumors [39,50]. We specifically examined the protein levels of ABCC1, ABCC2, ABCB1, and ABCG2 in exosomes, as they have been reported to contribute to drug-resistance activity in association with GH [29,32]. Additionally, we showed that the protein levels of ABCC1, ABCC2, ABCB1, and ABCG2 were elevated in presence of GH treatment in parent melanoma cells (Malme-3M and SK-MEL-28) (Appendix A).

GH significantly increased the levels of exosomal ABCC1 in the Malme-3M, SK-MEL-28, and SK-MEL-30 cells by 5-fold, 2.4-fold, and 7.2-fold, respectively, compared with the corresponding controls (Figure 2A,B, Table 1). Similarly significant increases were observed in exosomal ABCC1 following doxorubicin treatment in the Malme-3M (by 5.4-fold) and SK-MEL-28 (by 3.3-fold) cells. Furthermore, a significant increase in exosomal ABCC1 was observed in the Malme-3M (by 4.9-fold) and SK-MEL-28 (by 3.3-fold) cells following treatment with the GH and doxorubicin combination. Importantly, the addition of pegvisomant in combination with GH and doxorubicin rescued the exosomal ABCC1 levels back to those of the control. More evidently, when compared with GH treatment alone, pegvisomant combination treatment exhibited a significant reduction in exosomal ABCC1 from Malme-3M and SK-MEL-30 cells. Similar reductions were observed compared with doxorubicin treatment alone and in combination with GH. 

Regarding exosomal ABCB1, GH significantly increased the levels of exosomal ABCB1 from the Malme-3M (by 3.2-fold) cells as compared with the control (Figure 2A,D, Table 1), an effect that was also observed following doxorubicin stimulation in the Malme-3M cells (by 3.9-fold) as compared with the control. Furthermore, combined GH and doxorubicin treatment also resulted in a significant increase in exosomal ABCB1 in the Malme-3M cells (by 3.5-fold) as compared with the control. The addition of pegvisomant strikingly reduced the exosomal ABCB1 levels as compared with the GH and doxorubicin combination treatments. Related to exosomal ABCC2 and ABCG2, GH and doxorubicin increased, albeit not significantly, the levels in all the three cell lines as compared with the controls (Figure 2A,C,E, Table 1). An increasing trend was observed for the exosomal ABCC2 and ABCG2 levels following doxorubicin treatment in all three cell lines as compared with the control. The fold-changes are shown in Table 1. The original Western Blot data can be found in the Appendix A. These observations collectively suggest that GH and doxorubicin, alone or in combination, upregulate exosomal ABC transporters, while the addition of pegvisomant attenuates this increase.

#### 3.2.2. Effects of GH-Induced Melanoma-Derived Exosomes on Recipient Cells

Based on these findings, the action of GH promoting exosomal ABC transporters strongly suggests the potential role of GH in disseminating these transporters to other sensitive cancer cells via exosomes. Also, the above findings of reduced drug retention post administration with Exo^GH^ hint towards the GH-mediated exosomal transfer of ABC transporters. 

Our findings reveal a significant upregulation of ABCC1 levels in the Malme-3M recipient cells, following Exo^GH^ addition, exhibiting an approximate 2-fold increase compared with the control (Figure 2F,G, Table 1). Similarly, significant increases were induced by Exo^doxo^ administration in the Malme-3M cells, by 2-fold compared with the control. Furthermore, the ABCC1 transporter levels were significantly higher (1.6-fold) in the Malme-3M recipient cells compared with the control when Exo^doxo^ was administered. A significant reduction in ABCC1 was observed in the Malme-3M and SK-MEL-28 cells when Exo^GH+doxo+Peg^ was administered in the Malme-3M cells as compared with Exo^GH^. Similar reductions were observed when compared with Exo^doxo^ and Exo^GH+doxo^ administration in the Malme-3M cells. A similar, but not significant, increase in ABCB1 and ABCG2 was observed in the melanoma recipient cells when administered with Exo^GH^, Exo^doxo^, and Exo^GH+doxo^ as compared with the controls (Figure 2F,H,I, Table 1). The fold-changes are shown in Table 1. These observations suggest that ABC transporters are elevated in recipient cells when administered with exosomes from the GH- and doxorubicin-treated cells. whereas the ABC transporter levels are reduced when exposed to exosomes from the pegvisomant-treated melanoma cells treated in combination with GH and doxorubicin.

### 3.3. Blocking Autocrine/Paracrine GH Action Attenuates Exosomal ABC Transporter Levels

Pegvisomant caused a notable attenuation in the GH-mediated exosomal ABC transporter levels (discussed above). In some cases, the effect was even more pronounced, with exosomal ABC levels falling below those of the controls, indicating the suppression of autocrine/paracrine GH action. Extensive investigations by many groups have elucidated the autocrine role of GH in driving potent cancer properties, including the reduction in tumor sensitivity to chemotherapy [56,57,58]. In our previous study, we observed elevated levels of GH and GHR RNA in SK-MEL-28 melanoma cells following treatment with chemotherapies, including doxorubicin [59]. Interestingly, we detected basal levels of GH RNA, which were significantly downregulated when SK-MEL-28 and SK-MEL-30 were treated with pegvisomant (Appendix A). This downregulation was more pronounced in SK-MEL-30 when treated with both pegvisomant and doxorubicin.

To delineate the effects of autocrine/paracrine GH action on exosomal cargoes, we isolated exosomes from three melanoma cell lines: Malme-3M, SK-MEL-38, and SK-MEL30, treated with pegvisomant and doxorubicin. The exosomes are referred to as Exo^control^ (derived from PBS-treated cells), Exo^doxo^ (from doxorubicin-treated cells), Exo^Peg^ (from pegvisomant-treated cells), and Exo^Peg+doxo^ (from cells treated with pegvisomant and doxorubicin). Interestingly, the addition of pegvisomant did not affect the levels of exosomal ABCC1 and ABCB1 as compared with the controls (Figure 3A–C). As expected, a significant increase in exosomal ABCC1 and ABCB1 was observed when the melanoma cells were treated with doxorubicin alone, consistent with the findings described above, serving as a positive control. Remarkably, the addition of pegvisomant in combination with doxorubicin rescued the exosomal ABCC1 and ABCB1 levels back to those observed in the control groups. Hence, blocking autocrine/paracrine GH action inhibited doxorubicin-induced exosomal ABC transporters. The original Western Blot data can be found in the Appendix A.

### 3.4. Pegvisomant Treatment of Donor Melanoma Cells Attenuates Exosomal EMT-Inducing Effects

GH action plays a crucial role in regulating the migration of human tumors [60,61,62]. Multiple studies have shown that tumor-derived exosomes are involved in promoting migration [63,64]. To investigate the impact of GH-derived exosomes on the migration potential of melanoma cells, we utilized a wound-healing scratch assay [65]. 

Upon the administration of Exo^GH^ to the naïve melanoma recipient cells, SK-MEL-28 and SK-MEL-30, we observed a modest increase, though not significant, in migration by 10% and 5%, respectively, as compared with the controls (Figure 4A–D). Interestingly, treatment with Exo^doxo^ led to a significant acceleration in migration in the SK-MEL-30 (by 5%) cells compared with the controls (Figure 4D). Moreover, a notable enhancement in migration was observed in both the cell lines, with SK-MEL-30 showing a 5% increase and SK-MEL-28 exhibiting a more pronounced effect with a 25% rise, when incubated with Exo^GH+doxo^ (Figure 4B,D). Remarkably, the SK-MEL-28 cells incubated with Exo^GH+doxo^ demonstrated a significantly accelerated migration compared with the cells administered solely with Exo^GH^ (by 12%) and Exo^doxo^ (by 15%) (Figure 4B). The migration potential in both the cell lines was restored to normal when treated with Exo^GH+doxo+Peg^ as compared with the administration with Exo^GH+doxo^. 

The addition of GH and doxorubicin alone in donor cells accelerated the migration of SK-MEL-28 and SK-MEL-30 recipient cells via exosomes. This acceleration was further amplified in the presence of GH and doxorubicin together, showing an enhanced migration potential when combined, whereas pegvisomant dampened the effect on migration. These results indicate that GH and pegvisomant, when added to melanoma cells, alter the migration potential of the SK-MEL-28 and SK-MEL-30 recipient cells. 

### 3.5. GH Elevates the Expression of N-cadherin and MMP2 in Melamona-Derived Exosomes and Only Transfers N-cadherin to Recipient Cells

#### 3.5.1. Effects of GH on Cadherins and MMPs in Melanoma-Derived Exosomes

EMT is among the most well-known mechanisms underlying cancer cell migration/metastasis [66]. In addition, matrix metalloproteinases (MMPs), which are potent pro-EMT factors, modify the extracellular matrix of TME and increase the ability of cancer cells to migrate and infiltrate [67]. Studies have shown that GH serves as a potent driver of EMT in transformed cells [37,38] and has been shown to promote melanoma EMT in vitro and in vivo [32,38]. Studies have also shown the involvement of GH in elevating MMPs, particularly MMP2 and MMP9, thereby promoting cancer cell migration and invasion [57,68]. Additionally, we showed that the protein levels of N-cadherin and MMP2 are elevated in the presence of GH treatment in parent melanoma cells (Malme-3M and SK-MEL-28) (Appendix A). Extensive recent research has demonstrated that exosomes harbor these pro-EMT markers, ultimately elevating cancer cell migration and invasion [69,70,71,72]. To elucidate the potential of GH in modulating EMT and MMPs, we specifically examined the protein levels of N-cadherin, E-cadherin, MMP2, and MMP9, as they have been demonstrated to be modulated by GH [29,32].

GH, albeit not significantly, elevated the levels of exosomal N-cadherin in the Malme-3M and SK-MEL-30 cells as compared with the controls (Figure 5A,B). No difference was observed in the exosomal N-cadherin levels in the doxorubicin-induced cells. Notably, a significant surge in N-cadherin was observed in the Malme-3M cells (by 3.5-fold) when GH was combined with doxorubicin (Figure 5A,B, Table 2). Intriguingly, the addition of pegvisomant in the combination of GH and doxorubicin significantly reduced the exosomal N-cadherin levels, particularly in the Malme-3M cells when compared with GH and doxorubicin combination treatment. No change was observed in exosomal E-cadherin in the presence of GH or blocking the GH action (Appendix A). 

Similarly, GH elevated the exosomal MMP2 levels in all the cell lines, with a significant increase in the SK-MEL-30 cells (by 1.9-fold) as compared with the control (Figure 5A,C, Table 2). No difference was observed in the exosomal MMP2 levels in doxorubicin-induced cells. Furthermore, a significant increase in exosomal MMP2 was observed in the SK-MEL-30 cells (by 2-fold) when GH was added in combination with doxorubicin as compared with the control (Figure 4A,C). Pegvisomant addition to the aforementioned combination significantly attenuated the exosomal MMP2 in the SK-MEL-30 cells as compared with GH treatment alone and as compared with the combination of GH and doxorubicin combination treatment. No change was observed in exosomal MMP9 in the presence of GH or the blocking of the GH action (Appendix A). 

#### 3.5.2. Effects of GH-Induced Melanoma-Derived Exosomes on Cadherins and MMPs in Recipient Cells

The action of GH, particularly in the presence of doxorubicin, in promoting exosomal N-cadherin and MMP2, along with accelerated cell migration, indicates a potential role of GH in disseminating these proteins to treatment-naïve melanoma cells via exosomes. We observed that Exo^GH^ significantly augmented N-cadherin levels in the Malme-3M recipient cells (by 2.5-fold) and SK-MEL-28 (by 2-fold) as compared with the respective controls (Figure 5D,E, Table 2). Similarly, Exo^GH+doxo^ administration led to a significant increase in exosomal N-cadherin levels in the Malme-3M recipient cells (by 2-fold) as compared with the control cells. Notably, a significant reduction in N-cadherin was observed in the Malme-3M recipient cells when administered with Exo^GH+doxo+Peg^ compared with Exo^GH+doxo^. However, GH and doxorubicin alone and in combination demonstrated no change in the MMP2 levels in the recipient cells, whereas Exo^GH+doxo+Peg^ demonstrated a slight decrease in the MMP2 levels in SK-MEL-30 as compared with Exo^GH+doxo^ (Figure 5D,F, Table 2). The original Western Blot data can be found in the Appendix A.

In summary, our results demonstrated that GH upregulates the expression of exosomal N-cadherin and MMP2, while pegvisomant attenuates this GH-induced increase in exosomal N-cadherin and MMP2. Furthermore, exosome treatment of recipient cells suggests that GH can mediate the transfer of exosomal N-cadherin to the recipient cells, potentially aiding the transition to a mesenchymal phenotype, with pegvisomant mitigating this transfer. Conversely, no significant difference was observed in the MMP2 levels in any of the recipient cells. 

## 4. Discussion

Drug resistance continues to be a serious factor in the treatment of multiple cancers. Elevated ABC transporters in cancer cells, resulting in reduced intracellular drug accumulation, have been considered to be the major cause of chemotherapy resistance [73,74]. Moreover, EMT and elevated MMP levels also contribute to invasion, migration, metastasis, and chemoresistance. Extensive in vitro and in vivo studies have demonstrated that GH expression exacerbates chemotherapy resistance and cancer invasion in melanoma and other cancers via the upregulation of the levels of ABC transporters and EMT markers [29,32,75,76]. Conversely, these factors are significantly downregulated upon the inhibition of GHR, through GHR gene disruption (knock down) or pegvisomant. Several studies have highlighted the role of exosomes in disseminating these factors involved in drug resistance [39,50,77]. Therefore, we hypothesized that GH facilitates the packaging and transfer of these factors via exosomes, thereby modulating the drug efflux properties and migration of melanoma cells. In this study, we identify a novel role of GH in upregulating the packaging of multidrug efflux pumps and EMT markers in tumor-derived exosomes. Furthermore, we also demonstrate that GH-stimulated exosomal ABC transporters and EMT markers enhance the drug efflux activity and promote migration in the recipient cells. Notably, GHR antagonism attenuated the GH-stimulated exosomal ABC transporters and EMT markers. This consequently restored drug sensitivity and slowed cell migration. Additionally, we investigated the effects of blocking the autocrine/paracrine GH action, which highlighted the role of pegvisomant in attenuating the local GH-mediated exosomal ABC transporters.

Elevated ABC transporters in tumor cells, resulting in decreased intracellular drug accumulation, is a major reason for chemotherapy resistance in cancer. Exosomes are shown to transfer multiple ABC transporters from drug-resistant cells to drug-sensitive cells to upregulate drug resistance [78,79,80]. Our group has shown that GH elevates the specific ABC transporters, such as ABCC1, ABCC2, ABCB1, and ABCG2 in melanoma, hepatocellular carcinoma, and pancreatic cancer [32,49,76]. In the current study, we observed elevated levels of ABCC1 in exosomes from three melanoma cell lines stimulated with GH. Additionally, GH significantly increased the levels of exosomal ABCB1 from Malme-3M cells. The higher selectivity of GH packaging of ABCB1 in Malme-3M can be attributed to these cells having a heightened sensitivity to GH action, a correlation supported by the highest levels of GHR expression in Malme-3M cells compared with other melanoma cells within the NCI-60 panel, as previously reported by our laboratory [28]. The transfer of these ABC transporters via exosomes has been shown to promote chemoresistance in the recipient cells. Our finding reveals that GH-stimulated exosomes transferred high amounts of ABCC1 to the recipient cells in Malme3M, albeit less effectively in other cell lines. The elevated levels of ABC transporters in melanoma-derived exosomes could be a potential mechanism explaining the lower drug retention observed in recipient melanoma cells exposed to the GH-treated cells in our study. Additionally, we observed higher IC50 values in these recipient melanoma cells compared with the controls. Other mechanisms contributing to elevated drug retention and increased IC50 activity may involve the role of GH in modulating extracellular matrix (ECM) components such as collagen within the exosomes. GH is known to upregulate collagen synthesis [81]. A study in leukemia has demonstrated that collagen promotes doxorubicin resistance by reducing DNA damage through the inhibition of Tac1 activation [82]. Another study found that breast cancer cells grown on collagen I and fibronectin exhibited resistance to doxorubicin through upregulating ABCC1 [83]. Further, potential mechanisms include the role of GH in delivering non-coding RNAs involved in the elevation of drug retention and IC50 levels. The GH-regulated microRNA cluster 96-182-183 is known to promote EMT and invasion [44]. This cluster is involved in chemoresistance in multiple cancers [84,85,86]. These data, along with our findings, suggest that GH can not only provide a survival advantage to the cells within the chemotherapeutic milieu but can also promptly transfer the therapy resistance advantage to sensitive cancer cells.

In our study, we observed that doxorubicin treatment alone or in combination with GH increased the levels of specific exosomal ABC transporters. When these exosomes were exposed to recipient melanoma cells, we observed a decreased retention of the DiOC2(3) dye. This decrease in dye retention can be attributed to several potential mechanisms. Doxorubicin treatment has been found to upregulate the expression of various ATP-binding cassette (ABC) transporters in multiple cancers, including melanoma [87,88]. Another mechanism may involve the direct cytotoxic activity of doxorubicin present in exosomes. Studies in breast cancer have shown that doxorubicin-treated cells export the drug out of the cells via shed vesicles [89]. When these exosomes are taken up by the melanoma cells, doxorubicin can exert its cytotoxic effects directly within the recipient cells. Additionally, there might be competition for efflux pumps between the dye and doxorubicin. Doxorubicin is known to be a substrate for various efflux pumps, which actively transport drugs and other substances out of the cells [90]. When the melanoma cells are exposed to exosomes containing doxorubicin, the presence of doxorubicin can compete with DiOC2(3) for efflux via these transporters. This competition can lead to an increased efflux of DiOC2(3), resulting in a decreased retention of the dye within the cells. Further potential mechanisms include the modulation of non-coding RNA leading to chemoresistance. Exosomes from doxorubicin-resistant breast cancer cells have been shown to induce increased levels of miR-155 in doxorubicin-sensitive cells, thereby inducing chemoresistance [91]. This modulation of non-coding RNA can contribute to the decreased dye retention observed in the recipient melanoma cells.

Research from our laboratory and others has shown that inhibiting the GHR action sensitizes the human melanoma cells to chemotherapy by reducing the expression of ABC drug efflux pumps [29,31,32]. Here we report that inhibiting the GHR action via pegvisomant downregulates the levels of GH- and doxorubicin-induced exosomal ABCC1 in all the three cell lines. In addition, pegvisomant downregulates ABCC2, ABCB1, and ABCG2 in the Malme-3M cells. Strikingly, the transfer of ABCC1 to the Malme-3M recipient cells was even lower than in the controls, indicating the suppression of basal GH action. Mounting evidence has shown autocrine/paracrine GH expression within the microenvironment of multiple human cancers, including breast, endometrial, colon, liver, and prostate cancer [92,93,94]. Studies from Lobie’s group and others have shown that ‘forced’ autocrine GH expression is sufficient to promote the oncogenic transformation of mammary epithelial cells in vitro, whereas exogenously added hGH does not result in oncogenic transformation [57,95]. Autocrine/paracrine GH has also been implicated in resistance to radiation [93] and several chemotherapies, including mitomycin-C and daunorubicin [58,95]. Notably, doxorubicin has been shown to induce DNA damage, which in turn, induces GH production via p53 upregulation [30]. Moreover, our previous research has shown elevated levels of GH and GHR RNA in the SK-MEL-28 melanoma cells following treatment with chemotherapies, including doxorubicin [59]. Here we show that pegvisomant mitigates the doxorubicin-induced exosomal ABC transporters. Pegvisomant, thus, effectively downregulates the exosomal packaging of ABC transporters by suppressing basal autocrine/paracrine and potential chemotherapy-induced GH. Furthermore, our study reveals that exosomes obtained from melanoma cells treated with a combination of pegvisomant, GH, and doxorubicin significantly increased the retention of the DiOC2(3) dye and the reduction in IC50 in recipient melanoma cells compared with the recipient cells exposed to exosomes obtained from cells treated with GH or doxorubicin. These results can be attributed to pegvisomant blocking GHR, which likely downregulates the expression of the GH-induced factors, including ABC transporters, involved in elevated drug efflux. Consequently, this reduction in ABC transporter levels leads to a decreased efflux of DiOC2(3) and a reduction in IC50, thereby increasing their retention within the recipient cells.

EMT serves as a crucial initial process of metastasis. Exosomes from highly metastatic cells have been shown to ‘educate’ cells with a low metastatic potential to increase their migration and invasion [96]. Notably, these exosomes have been shown to elevate EMT by upregulating key markers, such as N-cadherin and vimentin [96]. Exosomal N-cadherin has been identified as a serum biomarker that is indicative of metastatic disease progression. Elevated levels of exosomal N-cadherin from serum have been detected in the serum of patients with osteosarcoma with pulmonary metastasis compared with those without metastasis [97]. Another study has shown that exosomes from late-stage lung cancer serum can induce migration, invasion, and proliferation in non-cancerous recipient human bronchial epithelial cells [98]. Our group has previously shown that GH increases both the protein and RNA levels of N-cadherin upon GH treatment, while GHR knockdown downregulates the N-cadherin expression in melanoma [38]. Here we report that GH treatment significantly elevated N-cadherin levels in melanoma-derived exosomes when treated in combination with doxorubicin. While no significant elevation in N-cadherin levels was seen in the GH-induced exosomes, a significant increase was observed when the GH-induced exosomes were administered in the recipient cells, particularly in Malme-3M and SK-MEL-28. Interestingly, similar levels of N-cadherin were maintained in the Malme-3M recipient cells when treated with exosomes from the GH and doxorubicin combination, whereas no such increase was seen in the recipient cells treated with exosomes from the doxorubicin-only treated cells. These results suggest that GH-induced exosomes potentially transfer factors that promote the expression of N-cadherin. Previous research has highlighted the role of various microRNAs, such as mir-19b-3p-derived from cancer stem cells, in the exosome-mediated transfer to renal cell carcinoma, resulting in the upregulation of N-cadherin via PTEN regulation [99]. Also, highly invasive gastric cancer has been shown to transfer lncRNA ZFAS1 to less invasive gastric cancer, thereby promoting migration and EMT through the upregulation of N-cadherin [100]. 

MMPs remodel ECM by breaking down collagen, thereby enhancing the aggressiveness of the cancer cells. MMP2 is shown to be carried by exosomes that are released by endothelial cells into the extracellular space, where MMP14 activates MMP2, facilitating ECM breakdown [101]. Studies have implicated the involvement of GH in MMP modulation, thus creating an invasive environment for the cancer cells. Lobie and group have shown an upregulation of MMP9 in endometrial cancer cells [69] and an upregulation of MMP2 and MMP9 in breast cancer cells when treated with GH [68]. In our study, we show that GH increases MMP2 in melanoma-derived exosomes. However, intriguingly, these elevated levels of exosomal MMP2 were not transferred to the recipient cells. This observation suggests that the GH-induced upregulation of exosomal MMP2 may contribute to the breakdown of ECM, thereby aiding the melanoma cell migration.

We also explored the expression levels of the evaluated ABC transporters, cadherins, and MMPs in TCGA datasets accessed via the UALCAN webserver. Our analysis revealed a significantly increased expression of ABCC1 and N-cadherin in metastatic melanoma compared with primary melanoma (Appendix A). Additionally, we evaluated the expression levels in other GH-sensitive cancers, including breast cancer and colon cancer. Our findings showed a significantly increased expression of ABCC1 in breast invasive carcinoma and colon adenocarcinoma compared with their respective normal samples (Appendix A). Furthermore, we observed a significant increase in N-cadherin expression in breast invasive carcinoma compared with the control samples (Appendix A). 

## 5. Conclusions

Our results demonstrate that excess GH upregulates ABC transporters, particularly ABCC1, and ABCB1 in the melanoma-derived exosomes, which in turn, can be transferred to treatment-naïve melanoma cells making them drug resistant. Additionally, GH stimulation leads to the upregulation of N-cadherin and MMP2 in melanoma-derived exosomes, with N-cadherin being transferred to treatment-naïve melanoma cells, enhancing their migratory potential. Although MMP2 is elevated in GH-induced exosomes, its transfer is not observed, indicating its predominant role in ECM degradation. Pegvisomant effectively inhibits the GH-induced upregulation of these factors in exosomes, consequently lowering the drug resistance and slowing down cell migration. Together, our results indicate a novel role of GH in disseminating the drug efflux and migration-promoting factors, contributing to chemotherapy resistance. Moreover, these factors hold promise as potential biomarkers in GH-sensitive cancers, aiding in therapy monitoring and the assessment of aggressive potential. Targeting the GHR with pegvisomant or other GH receptor antagonists presents a therapeutics strategy to mitigate the exosome-mediated dissemination of these factors, thereby overcoming chemotherapy resistance. 

## Figures and Tables

**Figure 1 cancers-16-02636-f001:**
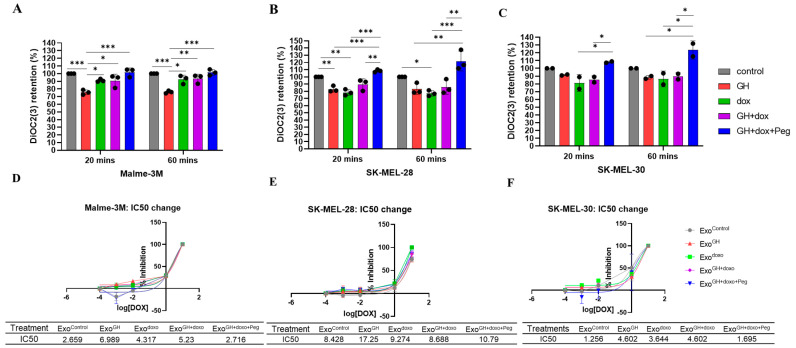
GHR antagonism in melanoma donor cells increased drug retention and decreased the IC50 of doxorubicin in recipient melanoma cells. A. Changes in the amounts of DiOC2(3) retained in the melanoma cells following administration with 20 μg/mL exosomes for 12-h, isolated from melanoma cells treated with 50 ng/mL GH independently or in combination with an IC50 dosage of doxorubicin and 500 nM pegvisomant. The fluorescent readouts from intracellular DiOC2(3) are presented for the Malme-3M (**A**), SK-MEL-28 (**B**), and SK-MEL-30 (**C**) melanoma cells. Three independent experiments were performed in triplicates and presented as the mean ± SD and *p* < 0.05 (*), *p* < 0.01 (**), and *p* < 0.001 (***). Changes in the IC50 values in Malme-3M (**D**), SK-MEL-28 (**E**), and SK-MEL-30 (**F**) melanoma cells following administration with the aforementioned exosomes and the increasing dosage of doxorubicin.

**Figure 2 cancers-16-02636-f002:**
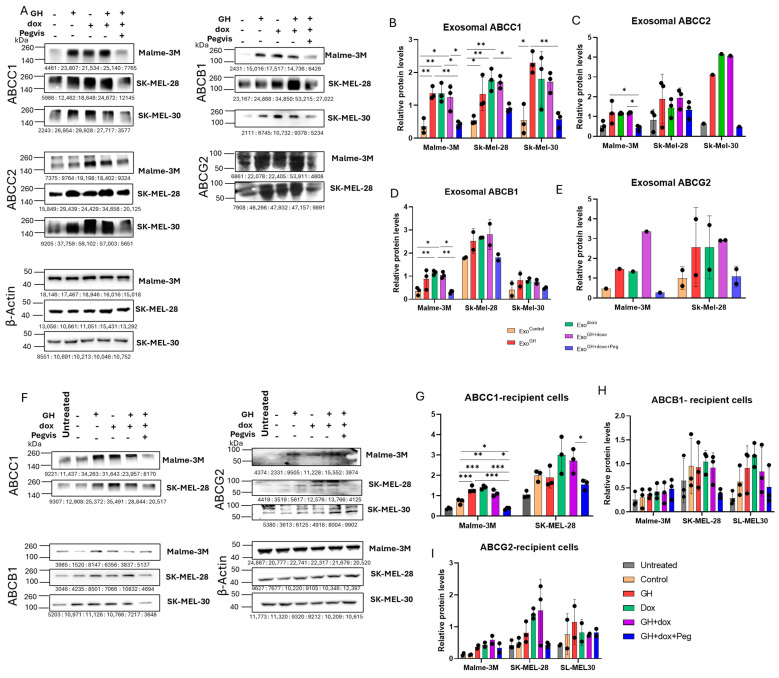
GH elevates ABC efflux pump levels in melanoma tumor-derived exosomes and corresponding recipient cells. (**A**) Protein levels of ABCC1, ABCC2, ABCB1, and ABCG2 in exosomes isolated from human melanoma cells, Malme-3M, SK-Mel-28, and SK-MEL30, 96-h post treatment with 50 ng/mL GH independently or in combination with EC50 dosage of doxorubicin, 500 nM pegvisomant. (**B**–**E**) Blots were quantified using ImageJ (version 1.8.0_345) and protein levels were normalized using β-actin as a control and presented as relative protein expression. Blots from the three independent experiments are presented as the mean ± SD and *p* < 0.05 (*), *p* < 0.01 (**), and *p* < 0.001 (***). (**F**) Protein levels of ABCC1, ACCB1, and ABCG2 in human melanoma cells: Malme-3M, SK-Mel-28, and SK-MEL30, treated with 20 μg/mL exosomes from abovementioned melanoma cells. (**G**–**I**) Blots were quantified using ImageJ and expressions were normalized against β-actin and presented as relative protein expression. Blots from the three independent experiments are presented as the mean ± SD and *p* < 0.05 (*), *p* < 0.01 (**), and *p* < 0.001 (***).

**Figure 3 cancers-16-02636-f003:**
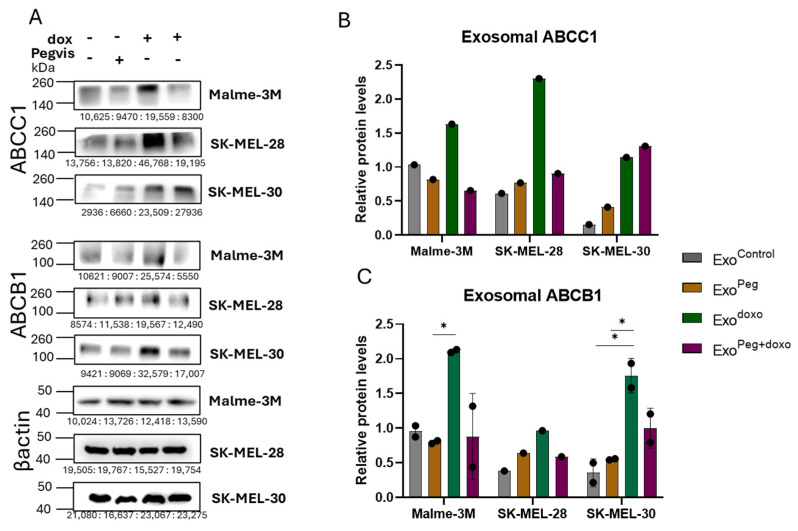
Blocking autocrine/paracrine GH action in melanoma suppresses ABC efflux pumps in tumor-derived exosomes. (**A**) Protein levels of ABCC1 and ABCB1 in exosomes isolated from human melanoma cells, Malme-3M, SK-Mel-28, and SK-MEL30, 96-h post treatment with 500 nM pegvisomant and an EC50 dosage of doxorubicin independently or in combination. (**B**,**C**) Blots were quantified using ImageJ (version 1.8.0_345) and expressions were normalized against β-actin and presented as relative protein expression. Blots from the two independent experiments are presented as the mean ± SD and *p* < 0.05 (*).

**Figure 4 cancers-16-02636-f004:**
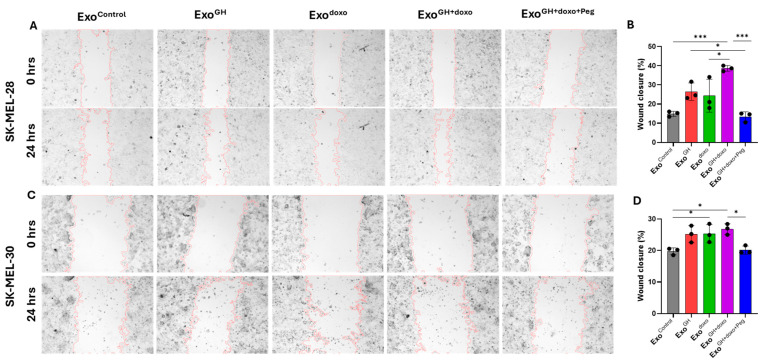
Pegvisomant attenuates exosome-induced cell migration in melanoma. Changes in wound closure following administration with 20 μg/mL exosomes, isolated from melanoma cells treated with 50 ng/mL GH independently or in combination with EC50 dosage of doxorubicin and 500 nM pegvisomant. Representative images for cell migration at the indicated time points are shown for SK-MEL-28 (**A**), SK-MEL-30 (**C**), and their respective quantifications (**B**,**D**). Three independent experiments were performed and presented as the mean ± SD and *p* < 0.05 (*), and *p* < 0.001 (***).

**Figure 5 cancers-16-02636-f005:**
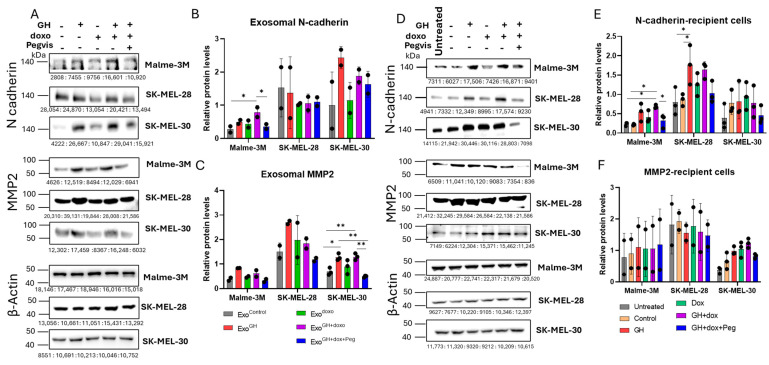
GH elevates N-cadherin and MMP2 in melanoma-derived exosomes and elevates N-cadherin in the corresponding recipient cells. (**A**) Protein levels of N-cadherin and MMP2 in exosomes isolated from human melanoma cells, Malme-3M, SK-Mel-28, and SK-MEL30, 96-h post treatment with 50 ng/mL GH independently or in combination with EC50 dosage of doxorubicin and 500 nM pegvisomant. (**B**,**C**) Blots were quantified using ImageJ (version 1.8.0_345) and expression was normalized using β-actin and presented as relative protein expression. Blots from the two independent experiments are presented as the mean ± SD and *p* < 0.05 (*), *p* < 0.01 (**), and *p* < 0.001 (***). (**D**) Protein levels of N-cadherin and MMP2 in human melanoma cells, Malme-3M, SK-Mel-28, and SK-MEL30, treated with 20 μg/mL exosomes from the aforementioned treated melanoma cells, (**E**,**F**) Blots were quantified using ImageJ and expression was normalized against β-actin and presented as relative protein expression. Blots from the three independent experiments are presented as the mean ± SD and *p* < 0.05 (*), and *p* < 0.01 (**).

**Table 1 cancers-16-02636-t001:** Average fold change in ABC transporter levels in exosomes and recipient cells.

ABC Transporter	Cell Lines	Exosomes				
		Control	GH	Doxo	Doxo + GH	Doxo + GH + Peg
ABCC1	Malme-3M	1.0	5.0	5.4	4.9	1.4
	SK-MEL-28	1.0	2.4	3.3	3.3	1.7
	SK-MEL-30	1.0	7.2	6.0	6.1	1.4
ABCC2	Malme-3M	1.0	2.8	2.4	2.5	0.8
	SK-MEL-28	1.0	2.2	2.5	3.4	2.3
ABCB1	SK-MEL-30	1.0	3.3	3.9	5.6	1.9
	Malme-3M	1.0	3.2	3.9	3.5	1.2
ABCG2	SK-MEL-28	1.0	1.4	1.5	1.6	1.0
	SK-MEL-30	1.0	2.7	3.2	2.9	1.9
		Recipient cells				
		Exo^Control^	Exo^GH^	Exo^Doxo^	Exo^Doxo+GH^	Exo^Doxo+GH+Peg^
ABCC1	Malme-3M	1.0	2.0	2.0	1.6	0.5
	SK-MEL-28	1.0	1.0	1.6	1.4	0.8
ABCB1	Malme-3M	1.0	1.5	1.8	1.5	0.7
	SK-MEL-28	1.0	1.0	1.3	1.1	0.5
	SK-MEL-30	1.0	1.5	2.1	1.5	0.9
ABCG2	Malme-3M	1.0	3.1	3.7	5.0	2.7
	SK-MEL-28	1.0	1.6	2.9	3.2	0.9
	SK-MEL-30	1.0	1.7	1.3	1.5	1.8

Average fold change is calculated based on the corresponding controls.

**Table 2 cancers-16-02636-t002:** Average fold change in MMP2 and N-cadherin levels in exosomes and recipient cells.

ABC Transporter	Cell Lines	Exosomes				
		Control	GH	Doxo	Doxo + GH	Doxo + GH + Peg
MMP2	Malme-3M	1.0	1.6	1.6	2.0	1.0
	SK-MEL-28	1.0	2.0	2.1	2.3	1.4
	SK-MEL-30	1.0	1.9	1.4	2.0	0.7
N-cadherin	Malme-3M	1.0	1.0	2.0	3.5	1.6
	SK-MEL-28	1.0	0.8	0.8	0.8	1.9
	SK-MEL-30	1.0	4.5	1.7	4.0	2.9
		Recipient Cells				
		Exo^Control^	Exo^GH^	Exo^Doxo^	Exo^Doxo+GH^	Exo^Doxo+GH+Peg^
MMP2	Malme-3M	1.0	1.2	1.1	1.0	0.8
	SK-MEL-28	1.0	0.8	1.9	0.8	0.8
	SK-MEL-30	1.0	1.6	1.8	2.0	1.4
N-cadherin	Malme-3M	1.0	2.5	1.9	3.1	1.5
	SK-MEL-28	1.0	2.0	1.4	1.9	1.2
	SK-MEL-30	1.0	1.0	1.2	1.0	0.7

Average fold change is calculated based on corresponding controls.

## Data Availability

The original contributions presented in the study are included in the article/Appendix A, further inquiries can be directed to the corresponding author(s).

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
