# Peer review of "Growth Hormone Upregulates Melanoma Drug Resistance and Migration via Melanoma-Derived Exosomes"

_cancers, 2024, doi:10.3390/cancers16152636_

Round 1

Reviewer 1 Report

Comments and Suggestions for Authors

Comments to the authors

Thank you for inviting me to evaluate the manuscript titled “Growth hormone regulates melanoma drug resistance and migration via melanoma derived exosomes”. In this study, Prateek Kulkarni and colleagues investigate the effects of growth hormone (GH) on exosomal cargo from melanoma cells and their role in drug resistance. Specifically, the authors identify that GH upregulates ABC transporters, increasing the drug efflux, contributing to chemotherapy resistance, and suggest that targeting the GH receptor with pegvisomant could overcome this resistance.

I have some comments as below:

Major points

1.       Kulkarni et al. evaluated the effects of exosomes derived from melanoma cells. The authors characterized the isolation of exosomes using NTA for exosome size and Western blot identifications for exosomes surface protein markers, such as CD63, CD9 and CD81. However, the International Society for Extracellular Vesicles (ISEV) recommends a more comprehensive approach to exosome characterization, including transmission electron microscopy (TEM) for morphology validation. I would still suggest that the authors include TEM analysis to provide this additional validation and support for the identification of exosome in their study.

2.      How and why did the authors choose a concentration of 20 ug/ml exosomes to treat the melanoma cells?

3.       For the drug retention assay, the number of independent experiments was too low. I suggest that the author perform an additional independent experiment to further confirm their results, achieving at least three independent experiments for statistical robustness, as done for the rest of the data presented. 

4.      Related to comment 2, the authors claim a 30% increase in drug retention after ExoGH+dox+Peg treatment in Malme-3M and SK-MEL28 cells. However, the data presented in Figure 1B and 1C (mean) do not reflect this change. The authors should review these data, including the additional experiment suggested above. 

5.       The authors do not directly demonstrate that selected dose of pegvisomant is working as a GHR antagonist. Could the authors provide supporting data for this?

6.      In page 5, lines 226-228, the authors state, “Addition of GH alone or in combination with doxorubicin induces cells to release 216 exosomes that relay a reduction in drug retention in recipient cells while pegvisomant treatment yields exosomes that elevate drug retention in recipient cells”. Which data or figure supports this statement?

7.       In Figure 2E and 2G, SK-Mel-30 cells are not represented. Could the authors clarify the reason for this omission? 

8.      In migration assay, SKMel-28 cells show significant increased migration capacity upon administration of ExoGH+doxo to recipient cells, and this effect is attenuated in the ExoGH+doxo+peg condition. However, the observed levels of N-cadherin and MMP2 proteins do not explained these results. Could the authors provide more convincing results or indicated other mechanisms involved in the restored drug sensitivity?

9.      Related to the previous comment, the authors propose higher levels of N-cadherin and MMP2 as biomarkers in GH-sensitive cancers, aiding in therapy-monitoring. Could they explore and confirm the potential of these biomarkers in the public available databases such as cBioPortal?

10.    In this study, the authors propose a role for GH in chemoresistance through its effect on migration. Have the authors explored the response of donor and recipient cells in other functional assays such as cell proliferation or invasion? It would be interesting to determine whether the regulation of GHR only affects migration or other cancer-related processes. Additionally, studying this regulation with other drugs besides doxorubicin may provide further insight into the restored drug sensitivity.

Minor points

1.       All figures should be improved as their current quality is too low, making the symbols representing statistical significance unreadable. In addition, the figures contain different symbols, and their meaning is not explained in the Material and Methods section or in the figure legends. Please improve the resolution of the figures and ensure that all meaningful symbols are clearly defined.

2.      Please check the X-axis in Figure 2C. The cell line names are missing, instead there are numbers.

3.       I suggest indicating the molecular weight of all bands represented in the Western blot images.

4.      The authors do not specify the recipient cells used in each experiment. Could you please specify this information?

5.       In Tables 1 and 2, the Fold change should be accompanied by the p-value and SD, which are essential to assess the reliability of the results.

6.      In Material and Methods section, could the authors specify the nominal molecular weight limit (NMWLof Amicon Ultra 15ml centrifugal filters used?

7.       Authors should check the abbreviation used for doxorubicin. Sometimes it appears as “doxo” and sometimes as “dox”, please unify the nomenclature throughout the manuscript.

8.      In supplementary data, the figure legends do not show the statistical information. Please review this and clarify whether the symbols should appear on the bars or if the data are not statistically significant.

Author Response

Major points

Comment 1: Kulkarni et al. evaluated the effects of exosomes derived from melanoma cells. The authors characterized the isolation of exosomes using NTA for exosome size and Western blot identifications for exosomes surface protein markers, such as CD63, CD9 and CD81. However, the International Society for Extracellular Vesicles (ISEV) recommends a more comprehensive approach to exosome characterization, including transmission electron microscopy (TEM) for morphology validation. I would still suggest that the authors include TEM analysis to provide this additional validation and support for the identification of exosome in their study.

Response 1: Thank you for your insightful comment. We appreciate your suggestion regarding the inclusion of transmission electron microscopy (TEM) for exosome characterization. We have indeed added TEM analysis of Malme-3M melanoma-derived exosomes, as shown in Supplementary Figure 2C. Additionally, we have included the detailed TEM protocol in the Materials and Methods section of our manuscript. Thank you for helping us improve the robustness of our study.

Comment 2: How and why did the authors choose a concentration of 20 ug/ml exosomes to treat the melanoma cells?

Response 2: In many studies within cancer research and stem cell research, exosome concentrations ranging from 10-40 µg/ml are commonly used for treatment administration. To support this, we have cited references 43, 44, and 45 as examples. In our preliminary experiments, we tested lower dosages of 5 µg/ml and 10 µg/ml but did not observe any significant effects in drug retention assay or migration assay (data not shown). Therefore, after optimizing the conditions, we chose a concentration of 20 µg/ml to proceed with our experiments.

Comment 3: For the drug retention assay, the number of independent experiments was too low. I suggest that the author perform an additional independent experiment to further confirm their results, achieving at least three independent experiments for statistical robustness, as done for the rest of the data presented. 

Response 3: We appreciate your suggestion regarding the number of independent experiments for the drug retention assay. We have performed additional independent experiments for two cell lines, Malme-3M and SK-MEL28, to achieve at the three replicates, ensuring statistical robustness. The updated data has been included in our revised manuscript. Thank you for helping us enhance the reliability of our findings.

Comment 4: Related to comment 2, the authors claim a 30% increase in drug retention after ExoGH+dox+Peg treatment in Malme-3M and SK-MEL28 cells. However, the data presented in Figure 1B and 1C (mean) do not reflect this change. The authors should review these data, including the additional experiment suggested above. 

Response 4. We appreciate your comments and concerns. We stand by our findings that demonstrate a notable enhancement in drug retention with ExoGH+dox+Peg treatment compared to ExoGH alone. Specifically, at the 20-minute time point in Malme-3M, the mean percentage of dioc2(3) retained was 75% with ExoGH treatment and 101% with ExoGH+dox+Peg treatment, resulting in a percentage increase of 36.6%. Similarly, for SK-MEL-28 cells at the same time point, the percentage increase was 31.71%. At the 60-minute time point, the percentage increase for Malme-3M cells was 34.21%, and for SK-MEL-28 cells, it was 46.99%.

We have carefully reviewed all data, including the additional experiments as suggested, and have ensured that the exact numbers are now presented in our revised manuscript. Thank you for guiding us towards improving the clarity and accuracy of our results.

Comment 5: The authors do not directly demonstrate that selected dose of pegvisomant is working as a GHR antagonist. Could the authors provide supporting data for this?

Response 5: We have taken your suggestion into consideration and have included the phosphorylation data of Y694/Y699 STAT5 as Supplementary Figure 1 for all three melanoma cell lines and also referenced it in the methods section in our manuscript. This phosphorylation of STAT5 serves as a standard to assess the functionality of GH and pegvisomant as GHR antagonists. Our results demonstrate that GH promotes STAT5 phosphorylation, while the GHR antagonist pegvisomant attenuates STAT5 phosphorylation levels at the selected dosages in the respective treatments. We believe this addition strengthens the support for our findings regarding the efficacy of pegvisomant as a GHR antagonist. Thank you for guiding us to enhance the clarity and completeness of our study.

Comment 6: In page 5, lines 226-228, the authors state, “Addition of GH alone or in combination with doxorubicin induces cells to release 216 exosomes that relay a reduction in drug retention in recipient cells while pegvisomant treatment yields exosomes that elevate drug retention in recipient cells”. Which data or figure supports this statement?

Response 6: Thank you for your insightful comment. This statement was intended as a summary of the drug retention assay results. However, we appreciate your note and have removed this statement from the specified section, as the detailed results are already discussed comprehensively in the discussion section.

Comment 7:Could the authors clarify the reason for this omission? 

Response 7: The SK-Mel-30 cell line was added later in the study, and it took additional time to optimize the antibody concentration and perform replicates. Therefore, we were unable to include the SK-Mel-30 data in Figures 2E and 2G. We appreciate your understanding and are working to include this data in future studies.

Comment 8: In migration assay, SKMel-28 cells show significant increased migration capacity upon administration of ExoGH+doxo to recipient cells, and this effect is attenuated in the ExoGH+doxo+peg condition. However, the observed levels of N-cadherin and MMP2 proteins do not explained these results. Could the authors provide more convincing results or indicated other mechanisms involved in the restored drug sensitivity?

Response 8: While the levels of N-cadherin and MMP2 proteins in SK-MEL-28 recipient cells did not fully explain the observations in migration assay leading to drug sensitivity, we believe that additional mechanisms may be involved.

  1. Role of tight junction proteins: GH and doxorubicin may promote the inclusion of other exosomal tight junction proteins that influence cell migration and drug sensitivity. Pegvisomant, by blocking GH action, could restore drug sensitivity. Research by Lobie’s group has shown that GH promotes cell migration and invasion by inhibiting the transcription of claudin-1, a key tight junction protein (Chen et al., 2017).
  2. Modulators of MMPs and cadherins: Other factors, such as WNT proteins delivered through GH and doxorubicin-induced exosomes, might affect migration capacity. For instance, autocrine GH has been demonstrated to stimulate WNT4 expression in breast cancer cells, which increases mesenchymal markers, including MMP2 and MMP7, thereby inducing cell migration (Vouyovitch et al., 2016).
  3. Role of non-coding RNA: GH may enhance the presence of specific non-coding RNAs within exosomes, which can modulate migration and restore drug sensitivity. GH has been shown to regulate the microRNA cluster 96-182-183, which is known to promote epithelial-mesenchymal transition (EMT) and invasion (Zhang et al., 2015).

Comment 9: Related to the previous comment, the authors propose higher levels of N-cadherin and MMP2 as biomarkers in GH-sensitive cancers, aiding in therapy-monitoring. Could they explore and confirm the potential of these biomarkers in the public available databases such as cBioPortal?

Response 9: We have explored the expression levels of N-cadherin and MMP2, as well as ABC transporters, using TCGA datasets accessed via the UALCAN webserver. For GH-sensitive cancers, we focused on melanoma, which is pertinent to our current study, and included breast cancer and colon cancer as additional GH-sensitive cancers based on a thorough literature review. Our analysis revealed a significantly increased expression of ABCC1 in metastatic melanoma compared to primary melanoma, as well as in breast invasive carcinoma and colon adenocarcinoma compared to their respective normal samples. Furthermore, we observed a significant increase in N-cadherin expression in metastatic melanoma compared to primary melanoma, and in breast invasive carcinoma compared to control samples. We have also added this in our discussion highlighted in yellow along with the data in supplementary figure 8.

These findings support the potential of N-cadherin and ABCC1 as biomarkers for monitoring GH-sensitive cancers, particularly in the context of metastatic disease. This analysis provides a robust foundation for further validation of these biomarkers in clinical settings.

Comment 10: In this study, the authors propose a role for GH in chemoresistance through its effect on migration. Have the authors explored the response of donor and recipient cells in other functional assays such as cell proliferation or invasion? It would be interesting to determine whether the regulation of GHR only affects migration or other cancer-related processes. Additionally, studying this regulation with other drugs besides doxorubicin may provide further insight into the restored drug sensitivity.

Response 10: Thank you for your insightful suggestions. We have indeed performed preliminary experiments on cell proliferation under the treatment conditions, but we did not observe significant changes. Therefore, we have decided to optimize the conditions further and plan to include comprehensive analyses of invasion and cell proliferation in our future studies. Additionally, we agree that exploring the regulation with other drugs besides doxorubicin may provide further insights into restored drug sensitivity, and we will consider this in our upcoming research. Thank you for your valuable feedback.

Minor points

Comment 1:  All figures should be improved as their current quality is too low, making the symbols representing statistical significance unreadable. In addition, the figures contain different symbols, and their meaning is not explained in the Material and Methods section or in the figure legends. Please improve the resolution of the figures and ensure that all meaningful symbols are clearly defined.

Response 1: We have ensured that the symbols representing statistical significance are now more uniform across all figures. Additionally, we have submitted the figures separately to the journal in a 600dpi format to improve their resolution. We apologize for the legibility issues in the pasted figures in the word format and appreciate your understanding. All meaningful symbols have been clearly defined in the revised figure legends.

Comment 2: Please check the X-axis in Figure 2C. The cell line names are missing, instead there are numbers.

Response 2: Thank you for pointing that out. We have corrected Figure 2C and have included the cell line names on the X-axis instead of numbers.

Comment 3: I suggest indicating the molecular weight of all bands represented in the Western blot images.

Response 3: Thank you for your suggestion. We have added the molecular weights and densitometry analyses to all bands represented in the Western blot images as requested by the journal. The updated figures have been included in the main text.

Comment 4: The authors do not specify the recipient cells used in each experiment. Could you please specify this information?

Response 4: We have included detailed information about the recipient cells used in each experiment wherever possible. In some instances, the term was used in a more general context for explanation purposes.

Comment 5: In Tables 1 and 2, the Fold change should be accompanied by the p-value and SD, which are essential to assess the reliability of the results.

Response 5: The numbers presented in Tables 1 and 2 represent the average fold change to provide a simplistic understanding of the data. The p-values and standard deviations (SD), which are essential for assessing the reliability of the results, are included in the corresponding figures where the detailed statistical analyses are presented.

Comment 6: In Material and Methods section, could the authors specify the nominal molecular weight limit (NMWL) of Amicon Ultra 15ml centrifugal filters used?

Response 6: We have used 100 kDa NMWL Amicon Ultra 15ml centrifugal filters and have added this information to the Material and Methods section.

Comment 7: Authors should check the abbreviation used for doxorubicin. Sometimes it appears as “doxo” and sometimes as “dox”, please unify the nomenclature throughout the manuscript.

Response 7: Thank you for your observation. We have standardized the abbreviation throughout the manuscript and have consistently used "doxo" for doxorubicin.

Comment 8:. In supplementary data, the figure legends do not show the statistical information. Please review this and clarify whether the symbols should appear on the bars or if the data are not statistically significant.

Response 8: We have reviewed the supplementary data and identified the missing statistical information in the figure legends. We have now updated the figures and their legends to include the appropriate statistical symbols and information wherever applicable.

Reviewer 2 Report

Comments and Suggestions for Authors

The manuscript was aimed to examine the effects of growth hormone in drug resistance of melanoma cells via melanoma-derived exosomes. In general the manuscript is well-prepared. 

I have the following suggestions and concerns anout this manuscript:

1) Given that growth hormone was previously shown to induce chemoresistance in a broad spectrum of human malignancies (as was also referred in this manuscript - lines 183-184), the authors have to explain why they observed the decreased retention of dye DiOC2(3) in all types of recipient melanoma cell lines, as was shown in Figure 1.  

2) What is the mechanism of the decreased dye retention in melanoma cells exposed to exosomes obtained from doxorubicin-treated cells? It might be due to the several mechanisms, including direct cytotoxic activity of doxorubicin present in the vesicles and/or competition in the efflux between doxorubicin and DiOC2(3) in melanoma cells, as well.

3) The authors have to clear the point for how long time the donor cells were exposed to doxorubicin before purification of the exosomes from them. It could be shown in Materials and Methods section or in the legend for Figure 1.  It's an important, since the cells have to up-regulate expression of ABC- transporters during doxorubicin exposure and this process requires  the extended culturing of cancer cells with chemotherapeutic agent.  Without this data the exosome-mediated transfer of ABC-transporters from recipient to donor cells is not explained (since all types of the recipient melanoma cells exhibited low basal levels of all types of ABC-transporters, as shown in Figure 2).  

4) The authors declare that they "found that blocking GH action with a drug called pegvisomant reduced the expression of these exosomal proteins, ultimately making cancer cells more responsive to chemotherapy.."  (lines 19-20), or "restoring drug sensitivity" (line 33 in abstract), etc.  However, manuscript lacks the evidences about how the aforementioned exosomes can affect the IC50 values for the chemotherapeutic agents, including doxorubicin, and therefore interfere with drug resistance. WB and or FACs data is also highly desirable to illustrate the changes in apoptotic markers (cleaved PARP, caspase-3 or Annexin-V positive cells, respectively) between the experimental conditions used for this study.  Lack of these evidences is one of the major concerns about this manuscript since it was the key point of this study ( as shown in the lines 89-90)  and the authors also highlighted it in the title of manuscript.

Minor:

1) the expresion of CD9 marker for SK-ML30 cells is missing -  as shown in the Supplementary Figure 2

Author Response

Comment 1: Given that growth hormone was previously shown to induce chemoresistance in a broad spectrum of human malignancies (as was also referred in this manuscript - lines 183-184), the authors have to explain why they observed the decreased retention of dye DiOC2(3) in all types of recipient melanoma cell lines, as was shown in Figure 1.  

Response 1: Thank you for your insightful comment. Our studies demonstrate that growth hormone (GH) reduces drug retention in recipient melanoma cells. We have highlighted the mechanisms involved, particularly the elevation of exosomal ABC transporters, which are transferred to recipient cells and subsequently promote the efflux of chemotherapy drugs, resulting in lower drug retention. Additionally, we have discussed further mechanisms in detail in the discussion section, which are highlighted in yellow. We have also further added explanations for the potential mechanisms for the change in DiOC2(3) retention across all types of recipient melanoma cell lines in the discussion section, marked in yellow for clarity. We appreciate your feedback, which has helped us improve the overall understanding of the manuscript.

Comment 2: What is the mechanism of the decreased dye retention in melanoma cells exposed to exosomes obtained from doxorubicin-treated cells? It might be due to the several mechanisms, including direct cytotoxic activity of doxorubicin present in the vesicles and/or competition in the efflux between doxorubicin and DiOC2(3) in melanoma cells, as well.

Response 2: The decreased dye retention in melanoma cells exposed to exosomes obtained from doxorubicin-treated cells can be attributed to multiple potential mechanisms two of which you pointed towards:

  1. Direct Cytotoxic Activity of Doxorubicin in Exosomes: Study on breast cancer have shown that doxorubicin-treated cells export the doxorubicin out of cells via shed vesicles (Shedden et al., 2003). When these exosomes are taken up by melanoma cells, the doxorubicin can potentially exert its cytotoxic effects directly within the recipient cells.
  2. Efflux Pump Competition: Doxorubicin is known to be a substrate for various efflux pumps, which actively transport drugs and other substances out of cells. When melanoma cells are exposed to exosomes containing doxorubicin, the presence of doxorubicin can compete with DiOC2(3) for efflux via these transporters. This competition can lead to increased efflux of DiOC2(3), resulting in decreased retention of the dye within the cells.
  3. Modulate non-coding RNA leading to chemoresistance: Exosomes from doxorubicin-resistant breast cancer cells have been shown to induce increased levels of miR-155 in doxorubicin-sensitive cells, thereby inducing chemoresistance (Santos et al., 2018). This modulation of non-coding RNA can contribute to the decreased dye retention observed in recipient melanoma cells.
  4. Modulation of Efflux Pump Expression or Activity: Doxorubicin treatment has been found to upregulate the expression of various ATP-binding cassette (ABC) transporters in multiple cancers, including melanoma (Calcagano et al., 2008). In our current study indicates that doxorubicin treated cells carry elevated level of ABCC1 and ABCB1, which in turn are delivered to recipient cells via exosomes, enhancing the efflux of DiOC2(3) and reducing its retention within the cells.

We have also added these mechanisms in the discussion section, which are highlighted in yellow.

Comment 3: The authors have to clear the point for how long time the donor cells were exposed to doxorubicin before purification of the exosomes from them. It could be shown in Materials and Methods section or in the legend for Figure 1.  It's an important, since the cells have to up-regulate expression of ABC- transporters during doxorubicin exposure and this process requires  the extended culturing of cancer cells with chemotherapeutic agent.  Without this data the exosome-mediated transfer of ABC-transporters from recipient to donor cells is not explained (since all types of the recipient melanoma cells exhibited low basal levels of all types of ABC-transporters, as shown in Figure 2).  

Response 3: The melanoma cells were treated with doxorubicin at the EC50 dosage for 96 hours, with replacement every 48 hours. This extended culturing period allowed for the upregulation of ABC transporters necessary for the exosome-mediated transfer of these transporters from donor to recipient cells. This information has been included in the Materials and Methods section.

Comment 4: The authors declare that they "found that blocking GH action with a drug called pegvisomant reduced the expression of these exosomal proteins, ultimately making cancer cells more responsive to chemotherapy.."  (lines 19-20), or "restoring drug sensitivity" (line 33 in abstract), etc.  However, manuscript lacks the evidences about how the aforementioned exosomes can affect the IC50 values for the chemotherapeutic agents, including doxorubicin, and therefore interfere with drug resistance. WB and or FACs data is also highly desirable to illustrate the changes in apoptotic markers (cleaved PARP, caspase-3 or Annexin-V positive cells, respectively) between the experimental conditions used for this study.  Lack of these evidences is one of the major concerns about this manuscript since it was the key point of this study ( as shown in the lines 89-90)  and the authors also highlighted it in the title of manuscript.

Response 4: We appreciate your feedback and understand the importance of providing comprehensive evidence on how the aforementioned exosomes affect IC50 values for chemotherapeutic agents. We examined IC50 values for doxorubicin following exosomes administration. We have demonstrated that the administration of exosomes derived from GH treated melanoma cells increased the IC50 values in all the recipient melanoma cells compared to the control. Conversely, the addition of pegvisomant restored drug sensitivity, reducing the IC50 values across these cell lines. This finding underscores the potential of pegvisomant to counteract GH-induced chemoresistance. We acknowledge that including data on apoptotic markers such as cleaved PARP, caspase-3, or Annexin-V positive cells would strengthen our conclusions. While this data is not currently included in our manuscript, we recognize its value and will prioritize this analysis in future studies to provide a more comprehensive understanding of the mechanisms at play.

Minor:

Comment 1: the expresion of CD9 marker for SK-ML30 cells is missing -  as shown in the Supplementary Figure 2

Response 1: Thank you for your observation. We have now added the CD9 expression data for SK-MEL-30 cells as shown in Supplementary Figure 2.

Reviewer 3 Report

Comments and Suggestions for Authors

This study explores how growth hormone (GH) contributes to this resistance. Researchers found that melanoma cells treated with GH release exosomes containing proteins that enhance drug resistance and cancer cell migration, especially when combined with the chemotherapy drug doxorubicin. Blocking GH action with pegvisomant reduced these proteins, making cancer cells more responsive to chemotherapy and less likely to migrate. This suggests that targeting GH could improve melanoma treatment by overcoming chemoresistance and inhibiting metastasis.

Important questions need to be addressed for publication, which is why it is considered that they need to carry out a major revision:

1. It is perplexing that, given the primary focus on the analysis of ABC transporter expression, the study does not include a broader range of markers, or at least melanoma-specific markers such as ABCB5. A comprehensive table of ABC transporters relevant to melanoma is available in the following publication, which is recommended for its specific insights into this type of cancer: Böhme I, Schönherr R, Eberle J, and Bosserhoff AK. (2021). Membrane transporters and channels in melanoma. Reviews of Physiology, Biochemistry and Pharmacology, 181, 269-374.

2. Why the authors in Figure 3 evaluate only ABCC1 and ABCB1 transporters. I again underline the need to increase the number of melanoma-specific markers and to maintain them throughout the study.

3. On page 11, lines 380-382 the authors refer to supplementary figure 5, but the data do not match. In the text they argue for the expression of n-cadherin and MMP2, but the figure shows ABC transporters. In addition, the sentence is written as if this result were a previous result from another work, if so, it makes no sense to include the figure, and if it is a new result, please rewrite it so that it is well understood.

4. On page 15, line 545: the ABC transporters are not written correctly, they are repeated.

Author Response

Comment 1: It is perplexing that, given the primary focus on the analysis of ABC transporter expression, the study does not include a broader range of markers, or at least melanoma-specific markers such as ABCB5. A comprehensive table of ABC transporters relevant to melanoma is available in the following publication, which is recommended for its specific insights into this type of cancer: Böhme I, Schönherr R, Eberle J, and Bosserhoff AK. (2021). Membrane transporters and channels in melanoma. Reviews of Physiology, Biochemistry and Pharmacology, 181, 269-374.

Response 1: Thank you for your valuable feedback and for pointing us towards the examination of a broader range of markers, including melanoma-specific markers such as ABCB5. We acknowledge the importance of including a comprehensive analysis of ABC transporters relevant to melanoma. In our study, we focused on the ABC transporters that have a well-established association with growth hormone, as demonstrated by our group and others. We appreciate your suggestion and will certainly explore a broader range of markers, including those highlighted in the recommended publication, in future studies.

Comment 2:. Why the authors in Figure 3 evaluate only ABCC1 and ABCB1 transporters. I again underline the need to increase the number of melanoma-specific markers and to maintain them throughout the study.

Response 2: We evaluated ABCC1 and ABCB1 in Figure 3 as we were building up our efforts to understand the autocrine and paracrine actions of growth hormone (GH). In the previous Figure 2, we observed that GH increased the levels of exosomal ABCC1 and ABCB1, and the addition of pegvisomant attenuated these ABC transporters. Therefore, we focused on checking the protein expression of these specific exosomal ABCC1 and ABCB1 under autocrine/paracrine GH conditions. We acknowledge the importance of including a wider range of melanoma-specific markers and will incorporate this in future studies.

Comment 3: On page 11, lines 380-382 the authors refer to supplementary figure 5, but the data do not match. In the text they argue for the expression of n-cadherin and MMP2, but the figure shows ABC transporters. In addition, the sentence is written as if this result were a previous result from another work, if so, it makes no sense to include the figure, and if it is a new result, please rewrite it so that it is well understood.

Response 3: Thank you for bringing this to our attention. We have cross-checked and confirmed that supplementary figure 5 indeed refers to the expression of N-cadherin and MMP2. We appreciate your observation regarding the clarity of the text, and we have revised the sentence to ensure it is grammatically correct and clearly conveys the intended message.

Comment 4:. On page 15, line 545: the ABC transporters are not written correctly, they are repeated.

Response 4: I’m unsure about the specific concern regarding the ABC transporters being repeated on page 15, line 545. I have reviewed that section and made sure there are no repetitions.

Round 2

Reviewer 1 Report

Comments and Suggestions for Authors

Dear Authors,

I greatly appreciate the effort and dedication you have put into following my recommendations. The manuscript has improved considerably with the changes you have made. However, I have noticed that some symbols representing statistical significance in the figures are still illegible. In addition, the figures contain different symbols, and their meanings are not explained in the “Materials and Methods” section or in the figure legends.

Please improve the resolution of the figures and ensure that all meaningful symbols are clearly defined. In particular, there is no indication of the difference between * and #.

Author Response

Comment: I greatly appreciate the effort and dedication you have put into following my recommendations. The manuscript has improved considerably with the changes you have made. However, I have noticed that some symbols representing statistical significance in the figures are still illegible. In addition, the figures contain different symbols, and their meanings are not explained in the “Materials and Methods” section or in the figure legends.

Please improve the resolution of the figures and ensure that all meaningful symbols are clearly defined. In particular, there is no indication of the difference between * and #.

Response: Thank you for your valuable feedback and for acknowledging the improvements made to the manuscript. We have increased the sizes of the symbols representing statistical significance to improve their legibility. Additionally, we have made the symbols consistent by using only '*', and we have removed other symbols to avoid any confusion. The meanings of these symbols are now clearly defined in both the “Materials and Methods” section and the figure legends.

Reviewer 2 Report

Comments and Suggestions for Authors

The authors responded to all my concerns and suggestions. The quality of revised manuscript was improved. The manuscript is suitable to publication in present form. 

Author Response

Comments: The authors responded to all my concerns and suggestions. The quality of revised manuscript was improved. The manuscript is suitable to publication in present form. 

Response: We sincerely appreciate your thorough review and valuable feedback, which significantly contributed to improving the quality of our manuscript. We are grateful for your time and effort in evaluating our work and are pleased to hear that the revised manuscript meets your standards for publication.

.

Reviewer 3 Report

Comments and Suggestions for Authors

The authors have responded satisfactorily to the requirements and have improved the manuscript considerably, making it suitable for publication.

Author Response

Comment:The authors have responded satisfactorily to the requirements and have improved the manuscript considerably, making it suitable for publication.

Response: We are grateful for your positive feedback and are pleased that our revisions have satisfactorily addressed your requirements. Your constructive comments have been invaluable in enhancing the quality of our manuscript.